# Association between the Static and Dynamic Lung Function and CT-Derived Thoracic Skeletal Muscle Measurements–A Retrospective Analysis of a 12-Month Observational Follow-Up Pilot Study

Mia Solholt Godthaab Brath [1,2,3,*], Sisse Dyrman Alsted [4], Marina Sahakyan [5], Esben Bolvig Mark [2,6], Jens Brøndum Frøkjær [2,5], Henrik Højgaard Rasmussen [2,7,8,9,10], Lasse Riis Østergaard [11], Rasmus Brath Christensen [5] and Ulla Møller Weinreich [1,2,3]

1.  Respiratory Research Aalborg (Reaal), Aalborg University Hospital, 9000 Aalborg, Denmark; ulw@rn.dk
2.  Department of Clinical Medicine, Aalborg University, 9220 Aalborg, Denmark; e.mark@rn.dk (E.B.M.); jebf@rn.dk (J.B.F.); hhr@rn.dk (H.H.R.)
3.  Department of Respiratory Diseases, Aalborg University Hospital, 9000 Aalborg, Denmark
4.  Department of General Medicine, North Region Hospital–Hjørring, 9800 Hjørring, Denmark; sisse.n@rn.dk
5.  Department of Radiology, Aalborg University Hospital, 9000 Aalborg, Denmark; m.sahakyan@rn.dk (M.S.); rach@rn.dk (R.B.C.)
6.  Mech-Sense, Department of Gastroenterology and Hepatology, Aalborg University Hospital, 9000 Aalborg, Denmark
7.  Department of Gastroenterology and Hepatology, Aalborg University Hospital, 9000 Aalborg, Denmark
8.  Danish Nutrition Science Center, Aalborg University Hospital, 9000 Aalborg, Denmark
9.  Center for Nutrition and Intestinal Failure, Aalborg University Hospital, 9000 Aalborg, Denmark
10. Department of Dietetic and Nutritional Research, Copenhagen University Hospitals, Herlev and Gentofte Hospitals, 2730 Herlev, Denmark
11. Medical Informatics Group, Department of Health Science and Technology, Aalborg University, 9220 Aalborg, Denmark; lasse@hst.aau.dk
*   Correspondence: m.brath@rn.dk

**Abstract:** Background: Patients with chronic obstructive pulmonary disease (COPD) with low skeletal muscle mass and severe airway obstruction have higher mortality risks. However, the relationship between dynamic/static lung function (LF) and thoracic skeletal muscle measurements (SMM) remains unclear. This study explored patient characteristics (weight, BMI, exacerbations, dynamic/static LF, sex differences in LF and SMM, and the link between LF and SMM changes). Methods: A retrospective analysis of a 12-month prospective follow-up study patients with stable COPD undergoing standardized treatment, covering mild to severe stages, was conducted. The baseline and follow-up assessments included computed tomography and body plethysmography. Results: This study included 35 patients (17 females and 18 males). This study revealed that females had more stable LF but tended to have greater declines in SMM areas and indices than males ($-5.4\%$ vs. $-1.9\%$, respectively), despite the fact that females were younger and had higher LF and less exacerbation than males. A multivariate linear regression showed a negative association between the inspiratory capacity/total lung capacity ratio (IC/TLC) and muscle fat area. Conclusions: The findings suggest distinct LF and BC progression patterns between male and female patients with COPD. A low IC/TLC ratio may predict increased muscle fat. Further studies are necessary to understand these relationships better.

**Keywords:** COPD; skeletal muscle; lung function; computed tomography; thorax

## 1. Introduction

Chronic obstructive pulmonary disease (COPD) is a complex progressive respiratory condition influenced by a combination of external and internal factors. Environmental

factors, such as air pollution [1,2], along with endogenous factors including genetics, inflammation, and oxidative stress, play significant roles in the development and progression of COPD [2]. This multifactorial etiology contributes to disease heterogeneity, affecting the structure and function of the respiratory system [2]. COPD is characterized by chronic airflow obstruction with significant systemic manifestations involving body composition. Computed tomography (CT), particularly chest high-resolution CT (HRCT), is an increasingly integrated part of the workup and management of COPD and marks a significant advancement in understanding and treating this multifaceted disease [3,4], especially in patients with COPD and frequent exacerbations, disproportionate severity of symptoms, significant airflow obstruction with hyperinflation, or those eligible for lung cancer screening [3].

CT scans offer detailed insights into pulmonary structures and pathology, including quantification of emphysema and air trapping [3,5,6]. In addition, they offer a window into the systemic manifestations of COPD, notably, changes in body composition [7–10], a factor increasingly acknowledged in COPD management [3]. Skeletal muscle alterations, such as low muscle mass or reductions in muscle mass, are linked to prolonged hospital admissions, an increased number of complications, and increased mortality [8,11,12].

The relationship between lung function and skeletal muscle characteristics in COPD patients is a multifaceted area affected by systemic inflammation, COPD exacerbations, and disease heterogeneity [13–16]. Systemic inflammation in COPD affects muscle and fat metabolism, leading to decreased muscle density and increased intra-muscular fat levels [17]. COPD exacerbations, characterized by acute worsening of respiratory symptoms, also significantly influence lung function and muscle health, often leading to reduced physical activity and heightened systemic inflammation [18]. The heterogeneous nature of COPD, with varied symptom patterns, disease progression, and response to treatment, further complicates our understanding of its systemic effects. CT scans provide a non-invasive means to evaluate these changes comprehensively.

Despite emerging evidence that links lung impairment with alterations in muscle measurements such as declines in the skeletal muscle area (SMA), index (SMI), and density (SMD) and increases in the intra-muscular fat area (IMFA), indices (IMFI), and density (IMFD) [19], the depth of this association in COPD patients remains under-explored. In particular, sexually dimorphic body composition differences between males and females, while well-established [20–25], are largely uncharted in patients with COPD.

We hypothesized that static and dynamic lung volumes could predict changes in thoracic muscle mass over 12 months in a cohort of patients with COPD. The aims of this study were as follows:

(1) To observe changes in demographic data, static and dynamic lung function parameters, and skeletal muscle over a 12 month period.

(2) To explore whether there are changes in body composition measurements including SMA, SMI, SMD, IMFA, and IMFD, as well as changes in dynamic (the forced expiratory volume in one second ($FEV_1$) and the $FEV_1$ to forced vital capacity ($FEV_1/FVC$) ratio) and static lung measures (total lung capacity (TLC), residual volume (RV), the inspiratory capacity to TLC ratio (IC/TLC), and the transfer factor for carbon monoxide (TLCO)).

(3) To investigate whether dynamic or static lung volumes at baseline are associated with changes in body composition measures after adjusting for sex, age, weight, height, and exacerbation frequency.

## 2. Materials and Methods

### 2.1. Study Design and Population

This was a retrospective analysis of a prospective observational 12-month follow-up pilot study from a previous study on Caucasian patients with COPD at the Respiratory Outpatient Clinic, Aalborg University Hospital, Denmark [26]. The baseline study was

conducted between April and August 2014 and follow-up was performed from May to September 2015.

The inclusion procedure was described in detail by Weinreich et al. [26]. In summary, the inclusion criteria were acceptable standards of the primary HR-CT scan (that is, the use of the correct HR-CT protocol, no technical issues such as motion artifacts, an acceptable field of view (FOV)), treatment of COPD according to GOLD recommendations at the time of follow-up [27], and no COPD exacerbations within six weeks of the study procedures at the time of follow-up. Re-scheduling of the procedures was possible; however, follow-up should be performed within four months of the one-year follow-up. The exclusion criteria were active cancer or suspected cancer and ongoing diagnostic workup for newly developed conditions that made the patient incapable of participating in study procedures (i.e., stroke, disability due to accident, major disability due to other intercurrent diseases). The number of exacerbations one year prior to baseline and in the year between the two CT scans was recorded. An exacerbation was defined according to concurrent GOLD recommendations [27]. Patients experiencing exacerbations were treated with a short-term course of 37.5 mg of prednisolone (10–14 days), either alone or in combination with antibiotics as per recommendations at the time of the study [27]. Some patients received less than 10–14 days due to side effects. Finally, technical issues with follow-up CT scans were excluded from the analysis.

### 2.2. Comorbidities

Comorbidity was defined as a condition/disease coexisting with COPD and was classified according to the International Classification of Disease tenth revision (ICD-10) [28].

All current comorbidities, based on the patient's medical records and prescriptions, were systematically recorded at baseline and follow-up by a trained healthcare professional and validated by an experienced senior consultant. Comorbidities are listed according to the International Classification of Diseases (ICD-10). Diseases/conditions that are self-limited/resolved have not been reported.

### 2.3. Lung Function

All patients underwent body plethysmography and a single breath diffusion examination (Jaeger Master Screen Body, Jaeger MS-PFT analyzer unit, LabManager V5.3.0) at baseline and follow-up. Trained personnel performed this procedure. The measures recorded were $FEV_1$, $FEV_1\%$, FVC, FVC%, $FEV_1/FVC$, IC, RV, TLC, and TLCO [29]. The device was calibrated daily before examination according to the manufacturer's specifications.

### 2.4. High-Resolution Computed Tomography

The patients underwent HR-CT at baseline and follow-up. HR-CT was conducted using GoldSeal Discovery CT750HD or LightSpeed Pro 32 (General Electric Healthcare, Chicago, IL, USA). All scanners were air calibrated daily, and constancy was tested monthly using producer-fabricated water air phantoms.

HRCT was performed with the patients in the supine position, arms raised above the head, and a full single-breath hold. The technical parameters used were a tube current of 120 kV, autoregulated mAs, single collimation of 0.625 mm, a scan field of view of 50.0 cm, pitch of 0.984, slice thickness of 0.625, and a kernel chest. Scans were conducted without an intravenous contrast medium. Three physicians assessed the HRCT; two senior radiologists with sub-specialization in chest CT and a senior pulmonologist independently assessed whether patients had bronchiectasis and emphysema.

### 2.5. Body Composition

Body composition was assessed based on a single axial slice at the first slice above the aortic arch on the HRCT [7,8]. Threshold-based semiautomated "Viking Slice" software (version 22.02.2021) was used for segmentation [30]. Attenuation between −29 and +150 HU was considered muscle tissue, while between −190 and −30 HU was considered

fat tissue [30,31]. Muscle and fat areas were reported as cm$^2$ and indices were reported as height-adjusted areas in cm$^2$/m$^2$, along with density in mean Hounsfield units (HUs). Figure 1 shows an example of segmentation.

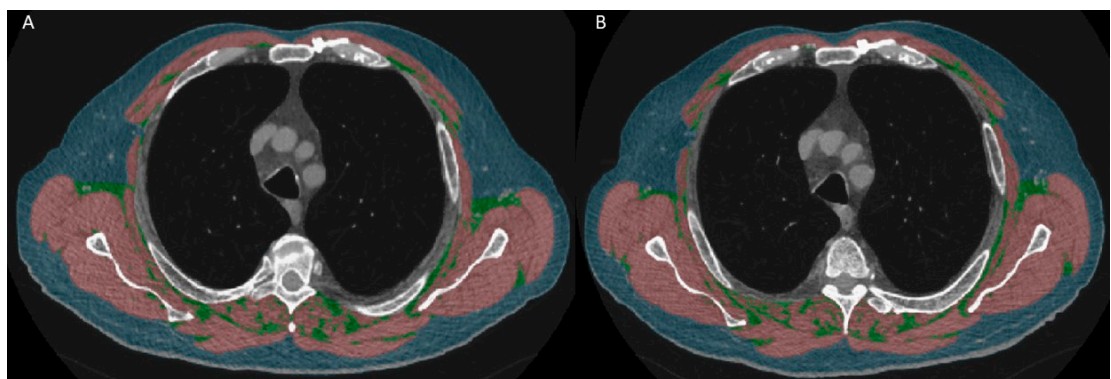

**Figure 1.** An example of body composition segmentation at the first slice above the aortic arch of a 70-year-old male patient at baseline (**A**) and follow-up (**B**). Red: skeletal muscle, blue: subcutaneous fat, green: inter- and intra-muscular fat.

Two trained reviewers assessed body composition using HRCT. Rater 1 was a pulmonologist with four years of experience and two years of experience and training in body composition analysis and chest CT. Rater 2 was a radiologist fellow with one year of experience in chest CT imaging and 12 years of experience as a resident.

*2.6. Statistical Analysis*

Demographic data were described by means or median values; normally distributed data were reported as the mean and 95% confidence interval (CI), whereas non-normally distributed data were reported as the median with minimum and maximum ranges (min; max). The Shapiro–Wilk test was used to assess normality. Categorical data were reported as proportions. If data were missing, the patient was excluded from the analysis.

For normally distributed data, an independent *t*-test comparing the sexes and a paired *t*-test were used to compare baseline and follow-up data. The Kruskal–Wallis test was used for non-normally distributed data.

Univariate linear regression analyses with robust variance were used to assess changes in body composition measures (SMA, SMI, SMD, IMFA, IMFI, and IMFD) adjusted for baseline characteristics such as sex, age, weight, height, body mass index (BMI), and exacerbation frequency per year. These analyses were performed independently for each characteristic. They also included the FEV$_1$, FEV$_1$/FVC ratio, total lung capacity (TLC), residual volume (RV), inspiratory capacity to TLC ratio (IC/TLC), and diffusing capacity for carbon monoxide (TLCO). Additionally, multivariate linear regressions were applied to predict changes in skeletal muscle and inter-muscular fat measurements, considering the dynamic and static lung volume parameters (TLC, RV, IC/TLC, and TLCO) and dynamic lung volumes (FEV$_1$ and FEV$_1$/FVC ratio), with adjustments for sex, age, weight, and height.

Multicollinearity and potential interactions between these covariates were assessed to ensure the validity of regression models.

A comparison between the included patients and patients excluded due to CT technical issues at follow-up was carried out on the demographic data and lung function measurements. The comparative analysis consisted of the chi square test, independent *t*-test, or Kruskal–Wallis test.

Statistical analyses were conducted using STATA 17.0 (Stata Corp LLC, College Station, TX, USA)

*2.7. Ethical Approval*

This study was approved by The Danish Local Science Ethics Committee N-20140019 and was conducted in accordance with the Helsinki Declaration. The data were registered and protected according to the Danish Data Protection Agency (F2023-062). All patients signed an informed consent form before enrolment and allowed data to be obtained from the Danish Shared Medical Record for further knowledge about overall health status and medication.

## 3. Results

At baseline, 111 patients were included. In total, 83 patients were deemed eligible and defined as the follow-up cohort [26]. The eligibility process for the follow-up cohort is shown in Figure 2. A total of 35 patients (17 female, 18 male) were included in the final study population. It is important to note that males in the study were older, taller, and heavier than the females, although the BMI values were similar in both groups.

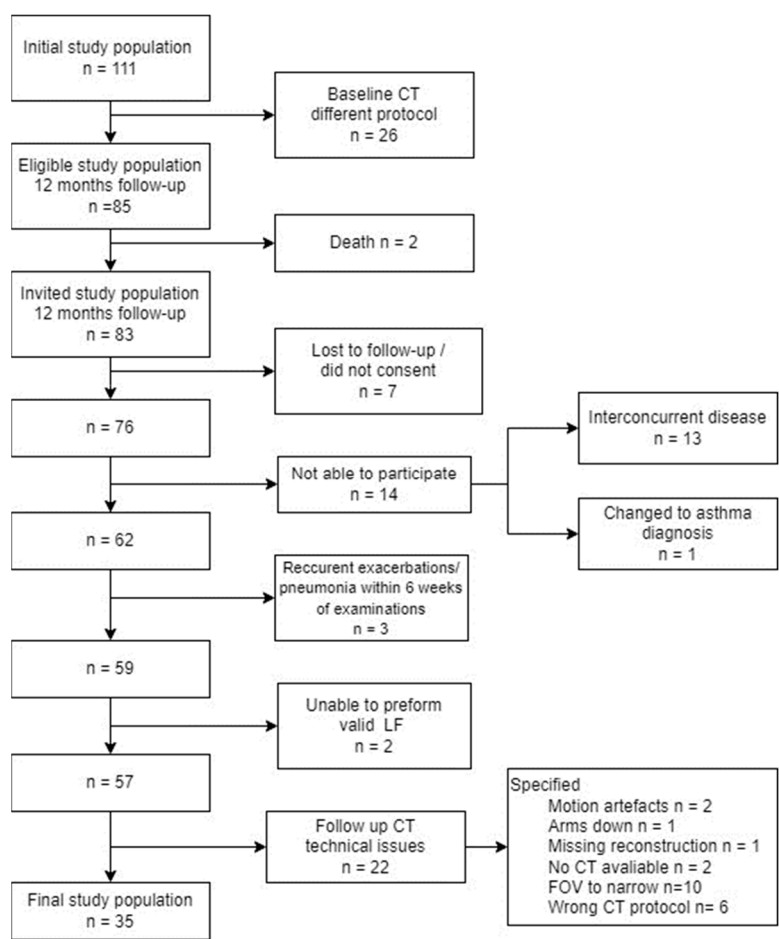

**Figure 2.** Flowchart of exclusion and inclusion of patients. CT: Computed tomography; FOV: field of view; LF: lung function assessment.

As seen in Table 1, the total study population suggests a potential trend towards a decrease in $FEV_1$, TLCO, and the number of exacerbations per year at follow-up compared to baseline, with RV showing a minor increase. Otherwise, most demographic data remained largely consistent between baseline and follow-up. For more details on the total study population, please refer to Appendix A Table A1a, and for changes within each sex from baseline to follow-up, please refer Table A1b.

As seen in Figure 3, all skeletal muscle measurements showed a trend toward a decrease, except IMFD, which significantly increased from baseline to follow-up. For a more detailed breakdown, please refer to the Appendix A material, Table A2.

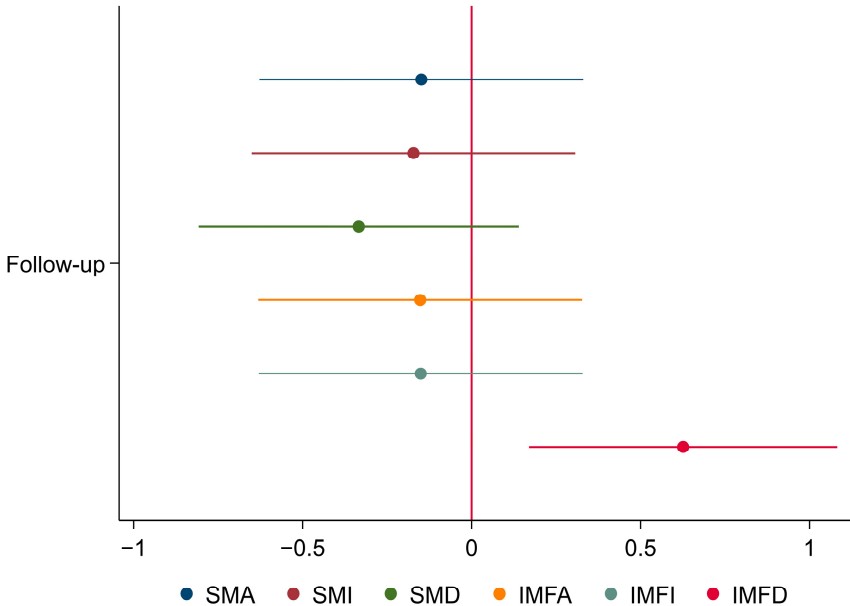

**Figure 3.** Forest plot of changes in thoracic muscle measurements at follow-up using standardized estimates and a 95% CI. The standardized estimates and CI were calculated by dividing the crude estimates by their respective standard deviation to facilitate meaningful comparison due to variations in effect size. CI: Confidence interval; SMA: skeletal muscle area; SMI: skeletal muscle index; SMD: skeletal muscle density; IMFA: inter-and intra-muscular fat area; IMFI: inter- and intra-muscular fat index; IMFD: inter-and intra-muscular fat density.

**Table 1.** Presents the characteristics of patients stratified by sex at baseline and after a 12-month follow-up period. The data are reported as means with 95% confidence intervals, except where otherwise specified. Non-normally distributed data are presented as medians with minimum and maximum ranges, and categorical data are presented as proportions.

| Patients Characteristics | Baseline | | | | 12-Month Follow-Up | | | |
|---|---|---|---|---|---|---|---|---|
| | Female | Male | Total | *p* | Female | Male | Total | *p* |
| Follow-up (months) median | | | | | | | 12 (10; 16) | |
| *n* (%) | 17 (48.6) | 18 (51.4) | 35 | | 17 (48.6) | 18 (51.4) | 35 | |
| Age (years) | 60.0 (55.2; 64.8) | 70.2 (67.4; 73.0) | | **0.00** | 61.2 (56.4; 66.0) | 71.2 (68.4; 74.0) | | **0.00** |
| Weight (kg) | 72.4 (63.3; 81.4) | 81.5 (73.6; 89.4) | | 0.14 | 71.7 (63.2; 80.3) | 82.3 (73.9; 90.8) | | 0.09 |
| Height (cm) | 166.6 (163.4; 169.9) | 171.4 (168.4; 174.4) | | **0.04** | 166.8 (163.7; 169.9) | 171.6 (168.3; 174.9) | | 0.05 |
| BMI (kg/m$^2$) | 26.0 (23.0; 29.0) | 27.6 (25.3; 29.9) | 26.8 (24.9; 28.7) | 0.41 | 25.8 (22.8; 28.8) | 27.8 (25.3; 30.3) | 26.8 (24.8; 28.8) | 0.32 |
| <18.5 (kg/m$^2$) | 1 (100.0) | 0 (0.0) | 1 | | 1 (100.0) | 0 (0.0) | 1 | |
| 18.5 to 24.9 (kg/m$^2$) | 9 (64.3) | 5 (35.7) | 14 | | 9 (60.0) | 6 (40.0) | 15 | |
| 25 to 29.9 (kg/m$^2$) | 2 (22.2) | 7 (77.8) | 9 | | 3 (33.3) | 6 (66.7) | 9 | |
| 30 to 34.9 (kg/m$^2$) | 3 (42.9) | 4 (57.1) | 7 | | 3 (42.9) | 4 (57.1) | 7 | |
| <35 (kg/m$^2$) | 2 (50.0) | 2 (50.0) | 4 | | 1 (33.3) | 2 (66.7) | 3 | |
| Smokers, *n* (%) | | | | | | | | |
| Current | 7 (58.3) | 5 (41.7) | 12 (54.5) | | 5 (50.0) | 5 (50.0) | 10 (45.5) | |
| Previous | 10 (43.5) | 13 (56.5) | 23 (47.9) | | 12 (48.0) | 13 (52.0) | 25 (52.1) | |
| Packyear, median | 30.0 (10.0; 53.0) | 40.0 (10.0; 88.0) | 35.0 (10.0; 88.0) | 0.06 | 31.5 (10.0; 55.0) | 43.0 (5.0; 80.0) | 35.0 (5.0; 80.0) | 0.13 |

**Table 1.** *Cont.*

| Patients Characteristics | Baseline | | | | 12-Month Follow-Up | | | |
|---|---|---|---|---|---|---|---|---|
| | Female | Male | Total | *p* | Female | Male | Total | *p* |
| **Lung function** | | | | | | | | |
| FEV$_1$ (%) | 67.8 (61.0; 74.5) | 60.7 (53.1; 68.3) | 64.1 (59.0; 69.3) | 0.19 | 67.1 (61.0; 73.2) | 58.3 (50.5; 66.1) | 62.6 (57.4; 67.7) | 0.09 |
| FEV$_1$/FVC-ratio | 57.4 (52.9; 61.9) | 53.7 (49.3; 58.2) | 55.5 (52.4; 58.7) | 0.26 | 57.2 (53.3; 61.1) | 53.0 (47.9; 58.1) | 55.1 (51.8; 58.3) | 0.21 |
| TLC % | 116.4 (109.5; 123.2) | 105.3 (97.2; 113.3) | 114.6 (67.6; 144.3) | 0.05 | 115.1 (107.7; 122.4) | 107.4 (99.4; 115.4) | 111.0 (78.8; 145.8) | 0.18 |
| IC % | 109.6 (98.3; 121.0) | 83.5 (73.4; 93.5) | 96.2 (87.5; 104.8) | **0.00** | 110.5 (100.8; 120.1) | 93.5 (85.0; 102.0) | 101.8 (94.8; 108.7) | **0.01** |
| IC/TLC % | 40.6 (35.4; 45.7) | 34.9 (30.0; 39.7) | 37.6 (34.2; 41.1) | 0.10 | 41.2 (37.0; 45.4) | 38.0 (33.6; 42.4) | 39.5 (36.6; 42.5) | 0.27 |
| RV % | 155.6 (135.4; 175.7) | 145.6 (125.2; 166.0) | 150.5 (136.2; 164.7) | 0.50 | 153.3 (136.0; 170.5) | 149.0 (129.7; 168.2) | 151.1 (138.3; 163.9) | 0.75 |
| TLCO % | 54.7 (47.2; 62.2) | 59.9 (50.6; 69.1) | 57.4 (51.4; 63.3) | 0.40 | 56.2 (47.9; 64.5) | 52.4 (46.8; 58.0) | 54.2 (49.3; 59.2) | 0.46 |
| **COPD-stage, *n*** | | | | | | | | |
| Mild | 3 | 2 | 5 | | 3 | 3 | 6 | |
| Moderate | 12 | 11 | 23 | | 13 | 8 | 21 | |
| Severe | 2 | 5 | 7 | | 1 | 7 | 8 | |
| mMRC, median | 1 (0; 3) | 1 (0; 3) | 1 (0; 3) | 0.45 | 1 (0; 2) | 1 (0; 4) | 1 (0; 4) | 0.49 |
| **COPD traits** | | | | | | | | |
| Exacerbations/year, median | 0 (0; 6) | 2 (0; 11) | 1.6 (0.9; 2.4) | **0.01** | 0 (0; 4) | 1 (0; 9) | 1.4 (0.7; 2.1) | 0.39 |
| ≥2 exacerbations/year | 3 (25.0) | 9 (75.0) | 12 (52.2) | | 4 (36.4) | 7 (63.6) | 11 (47.8) | |
| Emphysema, *n* (%) | 15 (48.4) | 16 (51.6) | 31 (49.2) | | 16 (50.0) | 16 (50.0) | 32 (50.8) | |
| **Type of comorbidities, *n* (%) \*** | | | | | | | | |
| Bronchiectasis | 10 (47.6) | 11 (52.4) | 21 (45.7) | | 12 (48.0) | 13 (52.0) | 25 (54.3) | |
| Hypertension | 6 (42.9) | 8 (57.1) | 14 (42.4) | | 8 (42.1) | 11 (57.9) | 19 (57.6) | |
| Hypercholesterolemia | 6 (60.0) | 4 (40.0) | 10 (38.5) | | 6 (54.5) | 5 (45.5) | 16 (61.5) | |
| Arthrosis | 5 (50.0) | 5 (50.0) | 10 (43.5) | | 7 (53.8) | 6 (46.2) | 13 (56.5) | |
| Osteoporosis | 6 (60.0) | 4 (40.0) | 10 (47.6) | | 6 (54.5) | 5 (45.5) | 11 (52.4) | |

*p*: Comparison between sexes; \*: the five most frequent comorbidities; BMI: body mass index; COPD: chronic obstructive pulmonary disease; TLCO: diffusing capacity of carbon monoxide in the lung; FEV$_1$: forced expiratory volume at one second; FVC: forced vital capacity; IC: inspiratory capacity; RV: residual volume; TLC: total lung capacity. Significant *p* values findings are marked in bold.

### 3.1. Follow-Up Stratified by Sex

As seen in Table 1, females reported fewer pack-years, had a larger IC, and generally experienced fewer exacerbations than males. Furthermore, there were notable changes after 12 months of follow-up in both the males and females. Males had trends towards an increased weight and BMI, whereas these measurements trended to decrease in females.

Regarding lung function, males experienced a trend towards a decline in FEV$_1$ and TLCO and a trend towards an increase in TLC, IC, and RV, whereas females' lung function parameters remained more similar from baseline to follow-up. Additionally, there was a decline in the number of males and females who experienced two or more exacerbations. For more details on the changes between baseline and follow-up for each sex, please see Table A1b.

As shown in Table 2, males had larger SMA, SMI, and IMFA values at baseline and follow-up than females. Females had a lower IMFD than males. Males exhibited a significant rise in IMFD, averaging an increase of 5.2 HU (3.3 to 7.0 HU), corresponding to a 6.9% increase. Additionally, females exhibited a larger decline in SMA, SMI, and SMD values than males, whereas males tended to decrease more in IMFA values and increased more in IMFD values than females. SMD increased similarly in both groups.

**Table 2.** Presents the skeletal muscle measurements and inter- and intra-muscular fat measures at baseline and the changes at follow-up for males and females. The data are reported as means with 95% confidence intervals.

| | Baseline | | | | Follow-Up | | | | |
|---|---|---|---|---|---|---|---|---|---|
| | Female | Male | p | | Female | Change in % | Male | Change in % | p |
| SMA (cm$^2$) | 139.4 (127.8; 150.9) | 193.5 (179.3; 207.8) | **<0.01** | SMA change (cm$^2$) | −7.5 (−10.3; −4.7) | −5.4 (−7.3; −3.4) | −4.1 (−7.4; −0.7) | −1.9 (−3.7; −0.1) | 0.13 |
| SMI (cm$^2$/m$^2$) | 50.1 (46.3; 53.9) | 65.7 (61.7; 69.6) | **<0.01** | SMI change (cm$^2$/m$^2$) | −2.7 (−3.7; −1.7) | −5.4 (−7.3; −3.4) | −1.3 (−2.4; −0.2) | −1.9 (−3.7; −0.1) | 0.08 |
| SMD (HU) | 39.1 (36.8; 41.5) | 39.9 (37.5; 42.4) | 0.64 | SMD change (HU) | −2.2 (−3.7; −0.6) | −5.1 (−9.5; −0.8) | −1.3 (−3.0; 0.4) | −3.1 (−7.0; 0.8) | 0.49 |
| IMFA (cm$^2$) | 23.4 (17.1; 29.7) | 28.4 (22.2; 34.5) | 0.28 | IMFA change (cm$^2$) | −1.5 (−4.1; 1.1) | −7.4 (−19.2; 4.4) | −2.4 (−5.9; 1.0) | −3.1 (−19.3; 13.2) | 0.67 |
| IMFI (cm$^2$/m$^2$) | 8.5 (6.2; 10.7) | 9.5 (7.5; 11.6) | 0.50 | IMFI change (cm$^2$/m$^2$) | −0.5 (−1.5; 0.4) | −7.4 (−19.2; 4.4) | −0.8 (−2.0; 0.4) | −3.1 (−19.3; 13.2) | 0.76 |
| IMFD (HU) | −77.1 (−80.1; −74.2) | −73.9 (−76.3; −71.4) | 0.10 | IMFD change (HU) | 3.6 (1.0; 6.1) | 4.7 (1.3; 8.1) | 5.2 (3.3; 7.0) * | 6.9 (4.4; 9.3) * | 0.32 |

*p*: *p*-value between sexes; * *p*: < 0.05 baseline compared to follow-up; IMFA: inter- and intra-muscular fat area; IMFI: inter- and intra-muscular fat index; IMFD: inter- and intra-muscular fat density; SMA: skeletal muscle area; SMI: skeletal muscle index; SMD: skeletal muscle density. Significant *p* values findings are marked in bold.

### 3.2. Unadjusted Linear Regression: 12-Month Changes in Thoracic Muscle Measurements

Univariate linear regression was employed to predict changes in thoracic muscle measurements (Δ) from baseline to follow-up from baseline demographic variables and lung function variables.

Of the demographic variables, age was positively associated with ΔSMA 0.24 (CI 0.04; 0.44, *p* = 0.02), ΔSMI 0.08 (CI 0.01; 0.15, *p* = 0.02), and ΔSMD 0.11 (CI 0.02; 0.19, *p* = 0.02) values. The same applies to the number of exacerbations and the ΔIMFD 0.61 (CI 0.13; 1.09, *p* = 0.01) values. Additionally, TLC was negatively associated with ΔSMD and ΔIMFI, whereas RV was positively associated with ΔIMFA and ΔIMFI. Finally, TLCO showed negative associations with ΔIMFA and ΔIMFI. The univariate regression is further specified in Appendix A Table A4. In analyzing the univariate regression models between each of the lung function parameters and thoracic muscle measurements, considerable variation in effect sizes was observed. To facilitate meaningful comparison, standardized estimates for each parameter were calculated. This approach involves dividing the crude estimate by its respective standard deviation, enabling a clear assessment of the impact of each parameter relative to the others.

Figure 4 presents the standardized estimates and confidence intervals of the univariate regressions. SMA and IMFA were excluded, as their values mirrored their respective indices. Although no model reached statistical significance, notable trends were observed. The TLC and IC/TLC ratio demonstrated the most pronounced negative impacts on ΔSMA and ΔSMI. Conversely, TLCO exhibited the most positive effect. In the case of SMD, TLC had the most negative effect, whereas the TLCO and FEV$_1$/FVC ratio had the most positive effects. For IMFA and IMFI, TLCO and IC/TLC showed the most negative effects, whereas RV and TLC displayed the most positive effects. The IC/TLC ratio also had the most negative effect on IMFD.

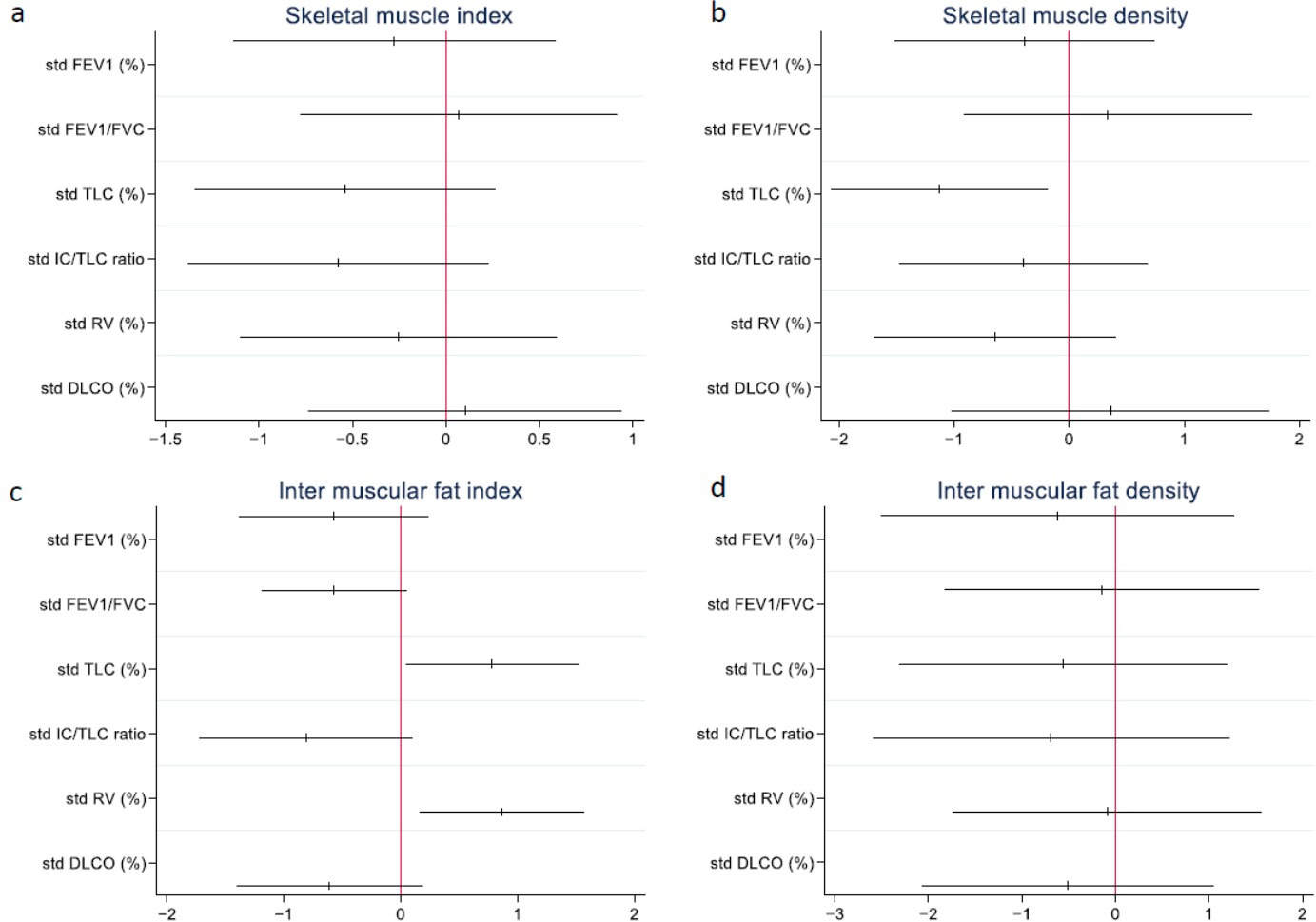

**Figure 4.** Forest plot between the difference in thoracic muscle measurements and lung function parameters using standardized estimates and a 95% CI. Standardized estimates and the CI were calculated by dividing with their respective standard deviation. (**a**) shows the skeletal muscle index and lung function parameters, (**b**) the skeletal muscle density and lung function parameters, (**c**) inter muscular fat and lung function parameters, and (**d**) intermuscular fat density and lung function parameters. CI: confidence interval; FEV$_1$: forced expiratory volume in one second; FVC: forced vital capacity; TLC: total lung capacity; IC/TLC: inspiratory capacity/TLC; RV: residual volume; TLCO: diffusing capacity for carbon monoxide.

### 3.3. Adjusted Linear Regression: Thoracic Muscle Measurements

Table 3 presents the results of multivariate linear regression models examining the relationship between various thoracic muscle measures and baseline lung function parameters, including the FEV$_1$, FEV$_1$/FVC ratio, TLC, IC/TLC ratio, RV, and TLCO. These models incorporate factors such as age, weight, height, sex, and the number of exacerbations. There were no signs of multicollinearity; however, there were significant interactions among some of the covariates in each model, which are specified in Appendix A Table A4.

Significant negative associations were observed between the IC/TLC ratio and both the difference in IMFA and IMFI. As seen in Figure 5, this suggests that higher IC/TLC ratios corresponded with reductions in IMFA and IMFI over a year. This effect was more pronounced in older participants, showing a significant interaction between the IC/TLC ratio and age.

**Table 3.** Presents the outcomes of the multivariate linear regression analysis, which evaluates the differences in skeletal muscle measures to each lung function parameter, with adjustments made for age, weight, height, sex, and the number of exacerbations. Data are presented as estimates and 95% confidence intervals.

| | | Estimate | | | Estimate |
|---|---|---|---|---|---|
| ΔSMA (cm$^2$) | FEV$_1$(%) | 0.004 (−0.18; 0.19, $p$ = 0.97) | Δ IMFA (cm$^2$) | FEV$_1$ (%) | −0.13 (−0.29; 0.03, $p$ = 0.11) |
| | FEV$_1$/FVC | 0.12 (−0.23; 0.47, $p$ = 0.47) | | FEV$_1$/FVC | −0.25 (−0.51; 0.02, $p$ = 0.07) |
| | TLC (%) | −0.04 (−0.20; 0.12, $p$ = 0.62) | | TLC (%) | 0.16 (−0.04; 0.36, $p$ = 0.12) |
| | IC/TLC | −4.15 (−36.99; 28.68, $p$ = 0.80) | | IC/TLC | −35.60 (−69.51; −1.70, ***p* = 0.04**) |
| | RV (%) | −0.002 (−0.06; 0.06, $p$ = 0.95) | | RV (%) | 0.06 (−0.01; 0.14, $p$ = 0.08) |
| | TLCO % | 0.04 (−0.11; 0.18, $p$ = 0.59) | | TLCO % | −0.10 (−0.25; 0.04, $p$ = 0.16) |
| ΔSMI (cm$^2$/m$^2$) | FEV$_1$ (%) | 0.00 (−0.06; 0.07, $p$ = 0.91) | Δ IMFI (cm$^2$/m$^2$) | FEV$_1$ (%) | −0.05 (−0.10; 0.01, $p$ = 0.10) |
| | FEV$_1$/FVC | 0.05 (−0.07; 0.17, $p$ = 0.43) | | FEV$_1$/FVC | −0.09 (−0.19; 0.01, $p$ = 0.07) |
| | TLC (%) | −0.01 (−0.07; 0.05. $p$ = 0.63) | | TLC (%) | 0.06 (−0.02; 0.13, $p$ = 0.12) |
| | IC/TLC | −1.32 (−12.78; 10.13, $p$ = 0.81 | | IC/TLC | −12.92 (−25.01; −0.83, ***p* = 0.04**) |
| | RV (%) | −0.00 (−0.02; 0.02, $p$ = 0.92) | | RV (%) | 0.02 (−0.00; 0.05, $p$ = 0.08) |
| | TLCO % | 0.02 (−0.04; 0.07, $p$ = 0.55) | | TLCO % | −0.04 (−0.09; 0.02, $p$ = 0.18) |
| ΔSMD (HU) | FEV$_1$ (%) | −0.01 (−0.09; 0.07, $p$ = 0.88) | Δ IMFD (HU) | FEV$_1$ (%) | −0.01 (−0.13; 0.12, $p$ = 0.90) |
| | FEV$_1$/FVC | 0.07 (−0.07; 0.20, $p$ = 0.31) | | FEV$_1$/FVC | 0.06 (−0.14; 0.26, $p$ = 0.56) |
| | TLC (%) | −0.06 (−0.13; 0.01, $p$ = 0.11) | | TLC (%) | −0.05 (−0.20; 0.10, $p$ = 0.48) |
| | IC/TLC | 0.05 (−13.40; 13.50, $p$ =0.99) | | IC/TLC | −0.81 (−22.312; 20.69, $p$ = 0.94) |
| | RV (%) | −0.01 (−0.04; 0.02, $p$ = 0.43) | | RV (%) | −0.01 (−0.07; 0.04, $p$ = 0.66) |
| | TLCO % | 0.04 (−0.04; 0.12, $p$ = 0.33) | | TLCO % | −0.02 (−0.10; 0.05, $p$ = 0.51) |

Δ: Difference between follow-up and baseline; BMI: body mass index; CI: confidence interval; TLCO: diffusing capacity of carbon monoxide; FEV$_1$: forced expiratory volume at one second; FVC: forced vital capacity; IC: inspiratory capacity; IMFA: inter- and intra-muscular fat area; IMFD: inter- and intra-muscular fat density; RV: residual volume; SMA: skeletal muscle area; SMD: skeletal muscle density; TLC: total lung capacity. Significant $p$ values findings are marked in bold.

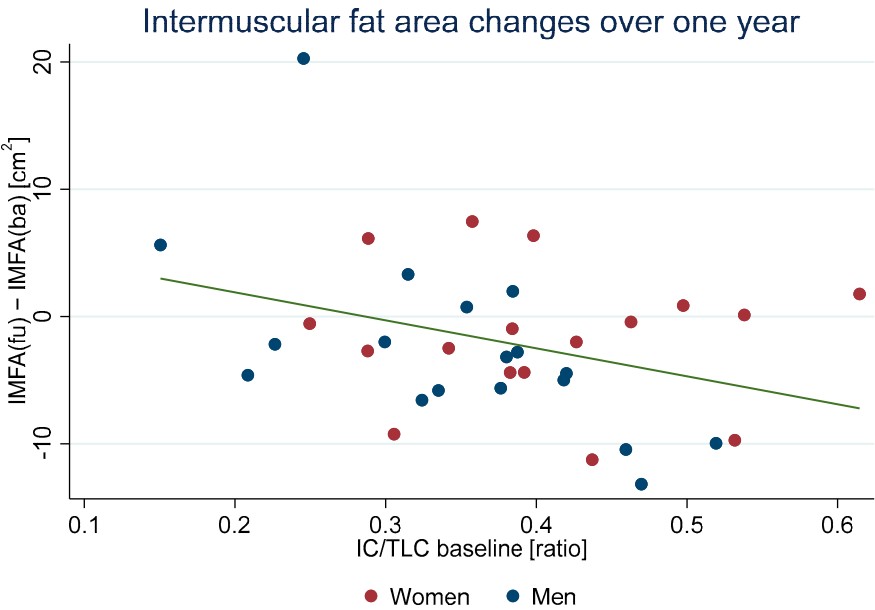

**Figure 5.** Scatterplot of the difference in IMFA and the baseline IC/TLC ratio. fu: Follow-up; ba: baseline; IMFA: inter-muscular fat area; IC/TLC: inspiratory capacity/total lung capacity.

*3.4. Waste Analysis*

A waste analysis was conducted comparing the 35 patients included in the study to the 22 patients excluded due to CT technical issues, all of whom had valid LF assessments at follow-up. The analysis revealed that the two groups were comparable across most of the variables. However, there was a significant difference in the IC/TLC ratio, with the excluded patients showing a lower mean ratio of 0.34 (CI 0.30; 0.38) compared to 0.40 (CI 0.37; 0.42) in the included patients ($p = 0.03$). A similar trend was observed in the TLCO among the excluded patients (45 (CI 37; 53) % vs. 54 (CI 49; 59) %, $p = 0.05$). Additionally, the RV was significantly larger in the excluded group, with a mean of 177% (CI 152; 203) compared to 151% (CI 138; 164) in the included group ($p = 0.04$). A similar trend among the excluded patients showed a higher median TLC of 127% (78; 147) compared to the TLC of 111% (CI 79; 146) in the included patients ($p = 0.06$). For more detailed information on waste analysis, please refer to Appendix A, Table A1c.

## 4. Discussion

This study explored changes in dynamic and static lung volumes and chest CT-based thoracic muscle measurements over a one-year follow-up period in patients with COPD. We observed trends towards a decline in thoracic muscle measurements, except for an increase in IMFD. While most lung function parameters remained stable, there were declining trends in TLC and TLCO, along with increasing trends in IC and IC/TLC. Notable sex-based differences were observed as females showed trends towards larger decreases in SMA and SMI, along with a smaller increase in IMFD compared to males. Despite the fact that females were younger, they had less severe COPD based on lung function parameters and had fewer exacerbations compared to males. In contrast, males exhibited an average increase in IC of 10%, along with a trend towards a decline in TLCO, whereas females remained stable. Multivariate linear regression revealed a significant negative association between IC/TLC at baseline and changes in IMFA and ΔIMFI over 12 months.

*4.1. A 12-Month Follow-Up: Lung Function and Thoracic Muscle Measurements*

In this study, we found trends towards increase in the IC and the IC/TLC ratio. This is intriguing and inconsistent with other longitudinal studies that have reported a decline in these measurements [32–35]. The IC/TLC ratio, a measurement of hyperinflation common in COPD [36–38], usually decreases as lung hyperinflation worsens owing to airway obstruction and loss of elastic recoil [38,39]. Hyperinflation results in a reduced IC and a worsened IC/TLC ratio, contributing to reduced exercise capacity, exacerbations, and increased risk of mortality [36–38,40–44]. A plausible explanation for the increase seen in our study could be the result of more effective control of lung hyperinflation attributed to rehabilitative measures combined with fewer exacerbations/years.

We observed trends towards a decline in skeletal muscle measurements, consistent with established research linking COPD and ageing to reduced muscle mass or function, increased fat infiltration, and conditions like muscle wasting, dysfunction, or sarcopenia [15,45–49]. These muscle changes are typically attributed to reduced muscle protein synthesis, hormonal alterations, and decreased physical activity [15,45–48]. Loss of muscle mass and strength has significant clinical implications, including reduced mobility, increased risk of exacerbations, and lowered quality of life [47,48].

Unexpectedly, our findings on muscle fat diverged from the prevailing trends noted in the existing literature, where increased muscle fat deposition is often associated with muscle dysfunction and sarcopenia in patients [15,50]. Contrary to these trends, we observed trends towards decreased inter- and intra-muscular fat area and indices over 12 months. This deviation prompts speculation on the potential factors for this discrepancy and its implications, as well as whether it represents an improvement or a sign of deterioration.

One intriguing hypothesis that arises from our observations is the possibility of tissue remodeling and fat compaction within the muscular structure. Tissue remodeling, a process characterized by structural and compositional alteration of muscle, could potentially lead to

a more compact fat distribution. This phenomenon could reflect an adaptive mechanism in response to chronic conditions or interventions aimed at preserving muscle function despite the presence of fat [51]. Fat compaction suggests reorganization of fat deposits, potentially leading to a decrease in the visible fat area, while possibly increasing the fat density.

In individuals with COPD, impaired adipose tissue function leads to accumulated fat buildup and systemic inflammation [52], which additionally hampers the regulation of lipid metabolism [53–56]. A similar phenomenon is also observed in athletes, who tend to have higher levels of intra-muscular fat, but maintain high insulin sensitivity [57,58] contrary to chronic conditions which are linked to increased insulin resistance [55]. In athletes the intra-muscular fat increase is thought to be an adaptive response, improving the efficiency of mitochondria within the muscle fibers, which in turn boosts fat metabolism and enhances insulin signaling [57,58].

The unique characteristics of the study population and variations in COPD severity offer a context for these findings. In more severe stages of COPD, patients often shift to a more catabolic state, leading to cachexia, characterized by weight, muscle, and fat loss [59]. However, our study participants were weight-stable and exhibited moderate COPD; therefore, cachectic deterioration seems unlikely. Instead, tissue remodeling and fat compaction may represent an adaptive response to maintain metabolic efficiency and muscle function during chronic disease progression [51].

Another aspect to consider is the potential impact of fewer exacerbations per year during follow-up. This reduction in exacerbations could limit fluctuations in inflammation and consequent impairment of lung function and physical activity. Additionally, this might also relate to the reduced usage of corticosteroids, which are known for their metabolic effects [60]. Alternatively, this decrease in fat measurements might be related to changes in dietary patterns or physical activity levels [47,61,62]; however, data on these aspects were unavailable.

### 4.2. Sex-Associated Changes in Lung Function and Skeletal Muscle Measurements

Intriguingly, in this study, males exhibited more pronounced increases in IC and a stronger trend towards increased IC/TLC ratio than females, despite being older and experiencing more exacerbations. Additionally, a trend towards an increase in RV among males suggests enhanced hyperinflation. The concurrent rise in IC by an average of 10%, alongside an increase in RV but only minimal changes in TLC, adds complexity to our understanding COPD's effect on lung volume and the reversibility of hyperinflation. This pattern might indicate a specific alteration in lung mechanics or volume distribution. The observed discrepancy could be due to a variety of factors, including the degree of air trapping, changes in chest wall compliance, or more effective management of hyperinflation in males.

These findings suggest distinct COPD progression patterns between sexes, which is consistent with the current literature [38,63]. This aligns with current differences in respiratory volumes and flows between sexes, which are well documented [38], as well as physiological and patient-reported outcomes, along with responses to treatment between males and females with COPD [63].

In contrast, females tended to have larger decrease in SMA, SMI, and SMD than males. These findings align with the recognized sexual dimorphic body composition [64], a concept that, while well-established, has been relatively unexplored in patients with COPD [65]. The prevalence of musculoskeletal issues, muscle wasting, and weakness is notably higher in females with COPD than in males [66,67]. These differences in maintaining muscle homeostasis are largely influenced by the sex hormones [64]. Additionally, males showed a trend towards reduction in IMFA, IMFI, and IMFD compared to females despite being older, having larger muscle areas, and more progressed COPD based on lung function parameters, including lower $FEV_1$ and TLCO. These findings add to the growing body of evidence on the complex interplay between sex-based differences and COPD progression, underlining the need for sex-specific approaches for both understanding and treating this condition.

### 4.3. Lung Function Parameters as a Predictor for Changes in Thoracic Muscle Measurements

Our study revealed a significant negative association between IC/TLC, ΔIMFA, and ΔIMFI over one year based on multivariate linear regression. This indicates that, with decreasing IC/TLC, there tends to be an increase in inter- and intra-muscular fat. This aligns with IC/TLC being a critical parameter for hyperinflation in patients with COPD, which again is linked to several metabolic changes, including increased energy expenditure and decreased exercise capacity, ultimately resulting in changes in body composition [36–38,41–44]. This suggests that the IC/TLC ratio may not only be an indicator of lung hyperinflation in COPD but also a potential marker for metabolic changes.

Recognizing changes is essential for effective COPD management, and should include addressing nutritional needs, managing comorbidities, and enhancing exercise tolerance and muscle strength [3]. Identifying patients with a low IC/TLC ratio could be key to determining those who might benefit from a more comprehensive treatment approach, such as pulmonary rehabilitation. Pulmonary rehabilitation, with its combination of exercise training, nutritional counseling, and patient education, is well-suited to address the complex challenges presented by these patients [68]. By targeting individuals with a low IC/TLC ratio for pulmonary rehabilitation, healthcare providers can offer tailored programs that potentially improve health outcomes and quality of life for those managing COPD.

### 4.4. Limitations

This pilot study included a small cohort of 35 patients with stable COPD from an outpatient clinic. Recruitment from an outpatient setting could mean that these patients represent a more severe and progressive disease state than that typically managed by general practitioners. However, the number of patients excluded due to intercurrent disease indicated a highly comorbid cohort. This scenario presents the risk of a two-sided exclusion bias, potentially overlooking both the most severely ill and healthiest patients due to resource limitations and patient interest.

It is important to note that many of our observations of sex-associated changes in lung function and skeletal muscle measurements did not yield any statistically significant results. This outcome may be due to the limited size of our study population, coupled with the brief duration of the follow-up period. Extending the follow-up period would offer a more comprehensive understanding of lung volumes and thoracic muscle measurements over time. Moreover, increasing the study population would not only enable the inclusion of a greater number of participants but also allow for further stratification of phenotypes and endotypes of COPD. This expansion would facilitate a more nuanced analysis, shedding light on the diverse effects of COPD across different patient groups. Additionally, introducing stratification based on sarcopenia or muscle wasting could enhance the analysis by allowing for a more detailed examination of these factors' impact on the study outcomes.

Furthermore, our waste analysis revealed that, for the included patients in the follow-up analysis, most parameters were comparable with those of patients excluded from follow-up. However, there was evidence that the excluded patients had a higher level of hyperinflation, which could potentially impact the study results. Including these patients might have allowed for more significant results, rather than mere trends.

As this was a retrospective study, new reconstructions of the CT scans could have expanded the study population. Unfortunately, many patients were excluded because of technical issues with the CT scans; notably, the displayed FOV was too narrow. The lack of raw data for new reconstruction further limited our ability to include more patients. Additionally, patients whose scans were conducted with the arms down were excluded, as the impact of arm positioning on skeletal muscle assessment using a single-slice approach remains unclear.

The primary phenotypic traits observed were the number of exacerbations and presence of emphysema. While most patients exhibit some degree of emphysema, those with this condition typically demonstrate more rapid and progressive changes in thoracic muscle measurements. There was also considerable variability in the exacerbation frequency per

year within the study group, although an overall reduction in exacerbations compared to baseline was noted. Future studies could be improved by including a broader range of COPD traits, such as chronic bronchitis, phlegm production, static and dynamic hyperinflation [42], and inflammatory and genetic markers, as seen in lung cancer studies, which have been associated with cancer cachexia [69] and include muscle strength (leg or hand grip) and exercise capacity.

A significant limitation relates to missing information regarding patients' participation in pulmonary or general rehabilitation in the study population. Rehabilitation may have affected the outcome of our study. Given the critical role of rehabilitation in managing COPD, including improving exercise capacity and quality of life [40,70], our study's insights into the efficacy of rehabilitation interventions are limited. This limitation is particularly relevant as rehabilitation practices can vary widely, and their effectiveness is closely linked to the specific methodologies employed and the individual patient's condition and response to treatment [40,70,71].

When considering the outcomes of this study, it is essential to approach them with caution, particularly concerning generalizability, as it may not be widely representative. However, this pilot study is a valuable step forward, shedding light on the progressive nature of lung function and thoracic muscle measurements and their potential interplay in patients with COPD. This finding underlines the importance of sexual dimorphism. To gain a more comprehensive understanding, more extensive research is necessary, considering aspects such as diet, exercise, sex-based hormonal influences, and genetic factors. Unraveling these relationships could provide crucial insights for the development of targeted interventions to improve respiratory and musculoskeletal health in patients with COPD.

## 5. Conclusions

This study indicated important sex-based differences in the progressive nature of COPD, including lung function parameters and thoracic muscle. Furthermore, IC/TLC may predict changes in inter-and intra-muscular fat. Focusing on these relationships in future research may provide a broader understanding of heterogeneity in COPD and could contribute to more tailored management and treatment.

**Author Contributions:** Conceptualization, M.S.G.B., S.D.A., M.S., E.B.M., J.B.F., H.H.R., L.R.Ø., R.B.C. and U.M.W.; methodology, M.S.G.B., S.D.A., J.B.F., E.B.M., H.H.R., L.R.Ø. and U.M.W.; software, E.B.M.; formal analysis, M.S.G.B.; investigation, M.S.G.B., M.S. and S.D.A.; resources, M.S.G.B.; data curation, M.S.G.B., S.D.A., M.S. and R.B.C.; writing—original draft preparation, M.S.G.B. and U.M.W.; writing—review and editing, M.S.G.B., S.D.A., M.S., E.B.M., J.B.F., H.H.R., L.R.Ø., R.B.C. and U.M.W.; visualization, M.S.G.B.; supervision, E.B.M., J.B.F., H.H.R., L.R.Ø. and U.M.W.; project administration, M.S.G.B. All authors have read and agreed to the published version of the manuscript.

**Funding:** This research received no external funding.

**Institutional Review Board Statement:** The study was conducted in accordance with the Declaration of Helsinki and approved by the North Regional Scientific Ethics Committee (approval code N-20140019, approval date 2 June 2015). Permission for data collection from patient medical records were obtained by North Region Denmark (approval code 2023-011707, approval date 13 April 2023) and The Danish Data Protection Agency (Journal number F2023-062, approval date 14 April 2023).

**Informed Consent Statement:** Informed consent was obtained from all patients involved in the study for initial data collection. For further medical record review in June 2023, we secured additional informed consent. For deceased patients, the North Region of Denmark waived the consent requirement (approval code 2023-011707, approval date 13 April 2023), allowing us to use their records.

**Data Availability Statement:** Data is not available due to ethical and legal data policies.

**Conflicts of Interest:** The authors declare no conflicts of interest.

## Appendix A

*Appendix A.1. Patient Characteristics for Total Populations*

**Table A1.** (**a**) Patient characteristics at baseline and at the 12-month follow-up. The data are reported as means with 95% confidence intervals, except where specified otherwise. Non-normal distributed data are presented as medians with the minimum and maximum range, and categorical data are presented as proportions. (**b**) Shows the patient characteristics divided by sex at baseline and at the 12-month follow-up. The data are reported as means with 95% confidence intervals, except where specified otherwise. Non-normal distributed data are presented as medians with the minimum, and maximum range and categorical data are presented as proportions. (**c**) Patient characteristics of excluded patients at the 12-month follow-up due to CT technical issues. The data are reported as means with 95% confidence intervals, except where specified otherwise. Non-normal distributed data are presented as medians with the minimum and maximum range and categorical data as proportions.

| (a) | | | |
|---|---|---|---|
| **Patient Characteristics** | **2014** | **2015** | ***p*-Values** |
| Study population, n | 35 | 35 | |
| Males | 18 | 18 | |
| Age (years), median | | | |
| Females | 61.0 (32.0; 75.0) | 62.0 (33.0; 76.0) | |
| Males | 69.0 (60.0; 82.0) | 70.0 (61.0; 83.0) | |
| Weight (kg) | | | |
| Females | 72.4 (63.3; 81.4) | 71.7 (63.2; 80.3) | |
| Males | 81.5 (73.6; 89.4) | 82.3 (73.9; 90.8) | |
| Height, (cm) | | | |
| Females | 166.6 (163.4; 169.9) | 166.8 (163.7; 169.9) | |
| Males | 171.4 (168.4; 174.4) | 171.6 (168.3; 174.9) | |
| BMI (kg/m$^2$) | 26.8 (24.9; 28.7) | 26.8 (24.8; 28.8) | 0.32 |
| Current smokers, n | 12 (54.5) | 10 (45.5) | |
| Previous smokers, n | 23 (47.9) | 25 (52.1) | 0.61 |
| Packyears, median | 35.0 (10.0; 88.0) | 35.0 (5.0; 80.0) | 0.69 |
| **Lung function** | | | |
| FEV$_1$ | 64.1 (59.0; 69.3) | 62.6 (57.4; 67.7) | 0.67 |
| FEV$_1$/FVC | 55.5 (52.4; 58.7) | 55.1 (51.8; 58.3) | 0.84 |
| TLC, median | 114.6 (67.6; 144.3) | 111.0 (78.8; 145.8) | 0.98 |
| IC | 96.2 (87.5; 104.8) | 101.8 (94.8; 108.7) | 0.33 |
| IC/TLC | 0.4 (0.3; 0.4) | 0.4 (0.4; 0.4) | 0.40 |
| RV (%) | 150.5 (136.2; 164.7) | 151.1 (138.3; 163.9) | 0.95 |
| TLCO (%) | 57.4 (51.4; 63.3) | 54.2 (49.3; 59.2) | 0.43 |
| mMRC, median | 1 (0; 3) | 1 (0; 4) | 0.82 |
| **COPD traits** | | | |
| Exacerbations/year, median | 1.6 (0.9; 2.4) | 1.4 (0.7; 2.1) | 0.66 |
| ≥2 exacerbations/year, n | 12 (52.2) | 11 (47.8) | |
| Emphysema, n | 31 (49.2) | 32 (50.8) | |
| **Type of comorbidities, n \*** | | | |
| Bronchiectasis | 21 (45.7) | 25 (54.3) | |
| Hypertension | 14 (42.4) | 19 (57.6) | |
| Hypercholesterolemia | 10 (38.5) | 16 (61.5) | |
| Arthrosis | 10 (43.5) | 13 (56.5) | |
| Osteoporosis | 10 (47.6) | 11 (52.4) | |

**Table A1.** *Cont.*

| (b) | | | | | | |
|---|---|---|---|---|---|---|
| | **Females** | | | **Males** | | |
| **Patient Characteristics** | **Baseline** | **Follow-Up** | *p* | **Baseline** | **Follow-Up** | *p* |
| n (%) | 17 (50.0) | 17 (50.0) | | 18 (50.0) | 18 (50.0) | |
| Age (years) | 61.0 (32.0; 75.0) | 62.0 (33.0; 76.0) | 0.41 | 70.2 (67.4; 73.0) | 71.2 (68.4; 74.0) | 0.62 |
| Weight (kg) | 72.4 (63.3; 81.4) | 71.7 (63.2; 80.3) | 0.92 | 81.5 (73.6; 89.4) | 82.3 (73.9; 90.8) | 0.89 |
| Height (cm) | 166.6 (163.4; 169.9) | 166.8 (163.7; 169.9) | 0.94 | 171.4 (168.4; 174.4) | 171.6 (168.3; 174.9) | 0.92 |
| BMI (kg/m$^2$) | 26.0 (23.0; 29.0) | 25.8 (22.8; 28.8) | 0.91 | 27.6 (25.3; 29.9) | 27.8 (25.3; 30.3) | 0.91 |
| <18.5 (kg/m$^2$) | 1 (50.0) | 1 (50.0) | | 0 (0.0) | 0 (0.0) | |
| 18.5 to 24.9 (kg/m$^2$) | 9 (50.0) | 9 (50.0) | | 5 (45.5) | 6 (54.5) | |
| 25 to 29.9 (kg/m$^2$) | 2 (40.0) | 3 (60.0) | | 7 (53.8) | 6 (46.2) | |
| 30 to 34.9 (kg/m$^2$) | 3 (50.0) | 3 (50.0) | | 4 (50.0) | 4 (50.0) | |
| <35 (kg/m$^2$) | 2 (66.7) | 1 (33.3) | | 2 (50.0) | 2 (50.0) | |
| **Smokers, n (%)** | | | | | | |
| Current | 7 (58.3) | 5 (41.7) | | 5 (50.0) | 5 (50.0) | |
| Previous | 10 (45.5) | 12 (54.5) | | 13 (50.0) | 13 (50.0) | |
| Packyear, median | 30.0 (10.0; 53.0) | 31.5 (10.0; 55.0) | | 40.0 (10.0; 88.0) | 43.0 (5.0; 80.0) | 0.92 |
| **Lung function** | | | | | | |
| FEV$_1$ (%) | 67.8 (61.0; 74.5) | 67.1 (61.0; 73.2) | 0.89 | 60.7 (53.1; 68.3) | 58.3 (50.5; 66.1) | 0.66 |
| FEV$_1$/FVC | 57.4 (52.9; 61.9) | 57.2 (53.3; 61.1) | 0.94 | 53.7 (49.3; 58.2) | 53.0 (47.9; 58.1) | 0.84 |
| TLC (%) | 117.8 (80.4; 144.3) | 116.3 (93.4; 145.8) | 0.64 | 105.3 (97.2; 113.3) | 107.4 (99.4; 115.4) | 0.72 |
| IC (%) | 109.6 (98.3; 121.0) | 110.5 (100.8; 120.1) | 0.92 | 83.5 (73.4; 93.5) | 93.5 (85.0; 102.0) | 0.14 |
| IC/TLC ratio | 0.4 (0.4; 0.5) | 0.4 (0.4; 0.5) | 0.84 | 0.3 (0.3; 0.4) | 0.4 (0.3; 0.4) | 0.32 |
| RV (%) | 155.6 (135.4; 175.7) | 153.3 (136.0; 170.5) | 0.87 | 145.6 (125.2; 166.0) | 149.0 (129.7; 168.2) | 0.82 |
| TLCO (%) | 54.7 (47.2; 62.2) | 56.2 (47.9; 64.5) | 0.80 | 59.9 (50.6; 69.1) | 52.4 (46.8; 58.0) | 0.18 |
| mMRC, median | 1 (0; 3) | 1 (0; 2) | 0.92 | 1 (0; 3) | 1 (0; 4) | 0.84 |
| **COPD traits** | | | | | | |
| Exacerbations/year, median | 0.0 (0.0; 6.0) | 0.0 (0.0; 4.0) | 0.61 | 1.5 (0.0; 11.0) | 1.0 (0.0; 9.0) | 0.31 |
| ≥2 exacerbations/year, n | 3 (42.9) | 4 (57.1) | | 9 (56.3) | 7 (43.8) | |
| Emphysema, n (%) | 15 (48.4) | 16 (51.6) | | 16 (50.0) | 16 (50.0) | |
| **Type of comorbidities, n *** | | | | | | |
| Bronchiectasis | 10 (45.5) | 12 (54.5) | | 11 (45.8) | 13 (54.2) | |
| Hypertension | 6 (42.9) | 8 (57.1) | | 8 (42.1) | 11 (57.9) | |
| Hypercholesterolemia | 4 (28.6) | 10 (71.4) | | 6 (50.0) | 6 (50.0) | |
| Arthrosis | 5 (41.7) | 7 (58.3) | | 5 (45.5) | 6 (54.5) | |
| Osteoporosis | 6 (50.0) | 6 (50.0) | | 4 (44.4) | 5 (55.6) | |
| (c) | | | | | | |
| **Patient Characteristics** | **2015 Included** | | **2015 Excluded** | | ***p*-Values** | |
| Study population | 35 | | 22 | | | |
| Males, n | 18 (47.4) | | 8 (36.4) | | 0.27 | |
| **Age (years), median** | | | | | | |
| Females | 62.0 (33.0; 76.0) | | 63.9 (60.6; 67.1) | | 0.39 | |
| Males | 70.0 (61.0; 83.0) | | 68.1 (58.5; 77.7) | | 0.43 | |
| **Weight (kg)** | | | | | | |
| Females | 71.7 (63.2; 80.3) | | 73.1 (64.6; 81.6) | | 0.82 | |
| Males | 82.3 (73.9; 90.8) | | 90.4 (73.0; 107.8) | | 0.37 | |
| **Height, (cm)** | | | | | | |

**Table A1.** *Cont.*

| | | | |
|---|---|---|---|
| Females | 166.8 (163.7; 169.9) | 166.8 (163.7; 169.9) | 0.36 |
| Males | 171.6 (168.3; 174.9) | 176.6 (171.6; 181.7) | 0.11 |
| BMI, (kg/m$^2$) | 26.8 (24.8; 28.8) | 27.8 (24.8; 30.8) | 0.57 |
| Current smokers, n | 10 (45.5) | 6 (27.3) | |
| Previous smokers, n | 25 (52.1) | 15 (68.2) | |
| Never smoker, n | 0 | 1 | |
| Packyears, median | 35.0 (5.0; 80.0) | 35.6 (15.0; 86.5) | 0.97 |
| **Lung function** | | | |
| FEV$_1$, | 62.6 (57.4; 67.7) | 60.2 (50.4; 70.0) | 0.64 |
| FEV$_1$/FVC | 55.1 (51.8; 58.3) | 54.4 (47.3; 61.6) | 0.86 |
| TLC, median | 111.0 (78.8; 145.8) | 126.6 (77.5; 146.7) | 0.06 |
| IC %, | 101.8 (94.8; 108.7) | 95.9 (84.0; 107.7) | 0.37 |
| IC/TLC | 0.40 (0.37; 0.42) | 0.34 (0.30; 0.38) | **0.03** |
| RV% | 151.1 (138.3; 163.9) | 177.8 (153.6; 202.0) | **0.04** |
| TLCO | 54.2 (49.3; 59.2) | 45.3 (37.8; 52.8) | 0.05 |
| mMRC, median | 1 (0; 4) | 2 (0; 4) | 0.23 |
| **COPD traits** | | | |
| Exacerbations/year, median | 1.4 (0.7; 2.1) | 1 (0; 8) | |
| ≥2 exacerbations/year, n | 11 (47.8) | 7 (31.8) | |
| Emphysema, n | 32 (50.8) | 17 (77.3) | |
| **Type of comorbidities, n *** | | | |
| Bronchiectasis | 25 (54.3) | 15 (68.2) | |
| Hypertension | 19 (57.6) | 13 (59.1) | |
| Hypercholesterolemia | 16 (61.5) | 12 (54.5) | |
| Arthrosis | 13 (56.5) | 12 (54.5) | |
| Osteoporosis | 11 (52.4) | 6 (27.3) | |

(**a**) *p*: Comparison between baseline and follow-up; *: the five most frequent comorbidities. (**b**) *p*: comparison between baseline and follow-up within the same sex; *: the five most frequent comorbidities. (**c**): *p*: comparison between included and excluded patients; *: the five most frequent comorbidities.

*Appendix A.2. Thoracic Muscle*

**Table A2.** Presents the thoracic skeletal muscle measurements for females and males at baseline and the follow-up.

| | Females | | | Males | | |
|---|---|---|---|---|---|---|
| | **Baseline** | **Follow-Up** | *p* | **Baseline** | **Follow-Up** | *p* |
| SMA (cm$^2$) | 139.4 (127.8; 150.9) | 131.9 (120.8; 142.9) | 0.36 | 193.5 (179.3; 207.8) | 189.5 (176.2; 202.7) | 0.68 |
| SMI (cm$^2$/m$^2$) | 50.1 (46.3; 53.9) | 47.5 (43.7; 51.2) | 0.33 | 65.7 (61.7; 69.6) | 64.4 (60.5; 68.2) | 0.64 |
| SMD (HU) | 39.1 (36.8; 41.5) | 37.0 (34.7; 39.2) | 0.21 | 39.9 (37.5; 42.4) | 38.6 (36.0; 41.2) | 0.47 |
| IMFA (cm$^2$) | 23.4 (17.1; 29.7) | 21.9 (16.0; 27.8) | 0.74 | 28.4 (22.2; 34.5) | 25.9 (19.8; 32.0) | 0.59 |
| IMFD (HU) | $-77.1$ ($-80.1$; $-74.2$) | $-73.6$ ($-77.7$; $-69.4$) | 0.17 | $-73.9$ ($-76.3$; $-71.4$) | $-68.7$ ($-71.1$; $-66.2$) | 0.01 |

*Appendix A.3. Univariate Linear Regression*

**Table A3.** Presents the results of the univariate linear regression analysis, which examines the relationship between the changes in skeletal muscle measures and each lung function parameter, as well as age, BMI, sex, and exacerbations.

| | ΔSMA (cm$^2$) | | | ΔIMFA | | |
|---|---|---|---|---|---|---|
| | **Estimate** | **CI** | | ***p*** | **Estimate** | **CI** | | ***p*** |
| FEV$_1$(%) | −0.05 | −0.21 | 0.11 | 0.51 | −0.10 | −0.24 | 0.05 | 0.19 |
| FEV$_1$/FVC | 0.02 | −0.24 | 0.28 | 0.86 | −0.16 | −0.34 | 0.02 | 0.08 |
| TLC (%) | −0.10 | −0.23 | 0.04 | 0.15 | 0.13 | 0.01 | 0.25 | **0.03** |
| IC/TLC ratio | −16.31 | −38.90 | 6.27 | 0.15 | −21.99 | −47.12 | 3.15 | 0.08 |
| RV (%) | −0.02 | −0.07 | 0.04 | 0.48 | 0.06 | 0.01 | 0.10 | **0.02** |
| TLCO % | 0.01 | −0.13 | 0.14 | 0.92 | −0.10 | −0.22 | 0.02 | 0.10 |
| Age (years) | 0.24 | 0.04 | 0.44 | **0.02** | −0.15 | −0.33 | 0.04 | 0.12 |
| Weight (kg) | −0.05 | −0.17 | 0.08 | 0.43 | −0.03 | −0.19 | 0.14 | 0.75 |
| Height (m) | −24.54 | −49.36 | 0.29 | 0.05 | −12.14 | −40.06 | 15.78 | 0.38 |
| Sex female vs. male | 3.46 | −1.08 | 8.01 | 0.13 | −0.94 | −5.38 | 3.50 | 0.67 |
| Exacerbations/year | 0.76 | −0.47 | 2.00 | 0.22 | 0.31 | −0.73 | 1.34 | 0.55 |

| | ΔSMI | | | | ΔIMFI | | |
|---|---|---|---|---|---|---|---|---|
| | **Estimate** | **CI** | | ***p*** | **Estimate** | **CI** | | ***p*** |
| FEV$_1$ (%) | −0.02 | −0.07 | 0.04 | 0.52 | −0.04 | −0.09 | 0.02 | 0.16 |
| FEV$_1$/FVC | 0.01 | −0.08 | 0.10 | 0.87 | −0.06 | −0.12 | 0.01 | 0.07 |
| TLC (%) | −0.03 | −0.08 | 0.02 | 0.18 | 0.05 | 0.00 | 0.09 | **0.04** |
| IC/TLC ratio | −5.69 | −13.62 | 2.25 | 0.15 | −7.99 | −16.98 | 0.99 | 0.08 |
| RV (%) | −0.01 | −0.03 | 0.01 | 0.54 | 0.02 | 0.00 | 0.04 | **0.02** |
| TLCO % | 0.01 | −0.04 | 0.05 | 0.81 | −0.03 | −0.08 | 0.01 | 0.13 |
| Age (years) | 0.08 | 0.01 | 0.15 | **0.02** | −0.05 | −0.11 | 0.02 | 0.14 |
| Weight (kg) | −0.01 | −0.05 | 0.03 | 0.58 | −0.01 | −0.07 | 0.05 | 0.80 |
| Height (m) | −5.94 | −14.75 | 2.87 | 0.18 | −3.51 | −13.72 | 6.71 | 0.49 |
| Sex female vs. male | 1.37 | −0.18 | 2.92 | 0.08 | −0.24 | −1.81 | 1.32 | 0.75 |
| Exacerbations/year | 0.28 | −0.13 | 0.69 | 0.18 | 0.10 | −0.25 | 0.46 | 0.56 |

| | ΔSMD | | | | ΔIMFD | | |
|---|---|---|---|---|---|---|---|---|
| | **Estimate** | **CI** | | ***p*** | **Estimate** | **CI** | | ***p*** |
| FEV$_1$(%) | −0.02 | −0.10 | 0.05 | 0.49 | −0.04 | −0.16 | 0.08 | 0.51 |
| FEV$_1$/FVC | 0.03 | −0.10 | 0.17 | 0.59 | −0.02 | −0.19 | 0.16 | 0.86 |
| TLC (%) | −0.07 | −0.12 | −0.01 | **0.02** | −0.03 | −0.14 | 0.07 | 0.52 |
| IC/TLC ratio | −3.94 | −14.55 | 6.67 | 0.46 | −6.79 | −25.65 | 12.07 | 0.47 |
| RV (%) | −0.02 | −0.04 | 0.01 | 0.22 | −0.00 | −0.04 | 0.04 | 0.91 |
| TLCO % | 0.02 | −0.06 | 0.10 | 0.60 | −0.03 | −0.11 | 0.06 | 0.51 |
| Age (years) | 0.11 | 0.02 | 0.19 | **0.02** | 0.05 | −0.16 | 0.26 | 0.66 |
| Weight (kg) | −0.00 | −0.08 | 0.08 | 0.95 | 0.02 | −0.07 | 0.12 | 0.62 |
| Height (m) | −6.99 | −27.68 | 13.70 | 0.50 | 12.98 | −13.06 | 39.02 | 0.32 |
| Sex female vs. male | 0.81 | −1.57 | 3.20 | 0.49 | 1.60 | −1.67 | 4.88 | 0.33 |
| Exacerbations/year | 0.28 | −0.18 | 0.74 | 0.22 | 0.61 | 0.13 | 1.09 | **0.01** |

Δ: The difference (follow-up–baseline); CI: confidence interval; BMI: body mass index; TLCO: diffusing capacity of carbon monoxide in the lung; FEV$_1$: forced expiratory volume at one second; FVC: forced vital capacity; IC: inspiratory capacity; IMFA: inter- and intra-muscular fat area; IMFD: inter- and intra-muscular fat density; RV: residual volume; SMA: skeletal muscle area; SMI: skeletal muscle index; SMD: skeletal muscle density; TLC: total lung capacity.

*Appendix A.4. Interaction Analysis*

**Table A4.** The interaction analysis for each of the multiple linear regressions. Only significant interactions are listed.

| | | Difference in SMA | Difference in SMI | Difference in SMD | Difference in IMFA | Difference in IMFI | Difference in IMFD |
|---|---|---|---|---|---|---|---|
| FEV$_1$ (%) | FEV$_1$##age | | | | $-0.02$ ($-0.05$; $-0.001$, $p = 0.039$) | $-0.01$ ($-0.02$; $-0.0004$, $p = 0.040$) | |
| | FEV$_1$##weight | | | | | | $-0.01$ ($-0.01$; $-0.002$, $p = 0.008$) |
| | Sex##weight | | | | | | 0.16 (0.01; 0.32, $p = 0.04$) |
| | Weight##exa | | | | | | 0.06 (0.01; 0.10, $p = 0.02$) |
| FEV$_1$/FVC | FEV$_1$/FVC##weight | | | | | | $-0.01$ ($-0.02$; $-0.002$, $p = 0.02$) |
| | Age##exa | 0.02 (0.001; 0.03, $p = 0.04$) | | | | | |
| | Sex##weight | | | | | | 0.17 (0.02; 0.32, $p = 0.03$) |
| | WeightI##exa | | | | | | 0.06 (0.01; 0,10, $p = 0.01$) |
| TLC (%) | TLC##weight | | | | 0.01 (0.002; 0014, $p = 0.01$) | 0.003 (0.001; 0.005, $p = 0.013$) | $-0.01$ ($-0.01$; $-0.0004$, $p = 0.04$) |
| | Sex##weight | | | | | | 0.17 (0.03; 0.31, $p =0.02$) |
| | Height##exa | | | | | | 16.4 (1.49; 31.3, $p = 0.03$) |
| | Weight##exa | | | | | | 0.06 (0.01; 0.10, $p = 0.01$) |
| IC/TLC | IC/TLC##age | | | | $-1.38$ ($-2.76$; $-0.004$, $p = 0.049$) | $-0.54$ ($-1.04$; $-0.04$, $p = 0.035$) | 1.68 (0.61; 2.76, $p = 0.003$) |

## represents the interaction analyses between two variables.

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
