# Peer review of "Association between the Static and Dynamic Lung Function and CT-Derived Thoracic Skeletal Muscle Measurements–A Retrospective Analysis of a 12-Month Observational Follow-Up Pilot Study"

_arm, doi:10.3390/arm92020015_

Round 1
Reviewer 1 Report
Comments and Suggestions for Authors
The present article is and interesting study about patient characteristics (weight, BMI, exacerbations, dynamic/static LF, sex differences in LF and skeletal muscle measurement (SMM), and the link between LF and SMM changes. The authors followed the rules of scientific sound and presentation throughout the manuscript (introduction, methods, results and discussion and conclusions). This study indicated important sex-based differences in the progressive nature of COPD, including lung function parameters and body composition. Furthermore, IC/TLC may predict changes in inter-and intramuscular fat. Focusing on these relationship, future research may provide a broader understanding of heterogeneity in COPD and could contribute to more tailored management and treatment options.
Reviewer 2 Report
Comments and Suggestions for Authors
Thank you for allowing me to review this pilot study, which explores the relationship between lung function parameters and thoracic skeletal muscle parameters (muscle and fat area and density) over a one-year follow-up period, in a monocentric retrospective design. A multivariate linear regression analysis evidences a negative association between baseline IC/TLC (a marker of static hyperinflation) and the change in inter- and intra-muscular fat area (whether normalized by body surface or not). In other words, the greater the hyperinflation, the greater the increase in muscular fat (a feature of sarcopenia) over time.
The topic is very interesting but the relatively small sample size associated to the heterogeneity of the disease prevents from drawing any definite conclusions; additionally, some data are somehow surprising (decrease in hyperinflation and muscular fat over time), although this is acknowledged by the authors.
Major comments :
- The term “body composition” is used throughout the manuscript but this would imply that a measurement of the whole body composition has been performed (e.g., by bioelectrical impedance analysis or dexametry). I suggest the authors replace it by “thoracic skeletal muscle area (± composition)” instead.
- Results, figure 2: there is a high number of excluded patients (22/35) due to follow-up CT technical issues. While this is correctly acknowledged and explained in the discussion, I wonder if this could have affected the results. Could the authors provide the demographics of these excluded patients and compare them to included one?
- I suggest the authors include a description of the baseline parameters at the beginning of the results section (this description is partly found later in the “follow-up stratified by sex” paragraph of the results section). In particular, the fact that the male patients are significantly older than the female patients is important and should be included at the beginning of the results.
- Table 1 description states that “As seen in Table 1, the total study population had a decrease in FEV1, DLCO, and the number of exacerbation/year at follow-up compared to baseline, while RV showed a slight increase. Otherwise, most demographic data were similar…”. However, these conclusions, which are supported by Table A1 and not Table 1, should be toned down as none of these parameters change significantly. At best, there is a trend but this is not the case for FEV1 and DLCO for example. I suggest the authors re-write this paragraph. If a parameter demonstrates an interesting trend over time, a graph could help to highlight it.
- Results section, in the paragraph “follow-up stratified by sex”: the authors should distinguish what is statistically significant from what is a trend in this paragraph. I also suggest re-writing. For example, it is stated that “regarding lung function, male experienced a decline in FEV1 and DLCO” but this is not supported by the presented data.
- Male patients exhibit over the follow-up period an increase in the inspiratory capacity as well as an increase in the residual volume; this should result in a marked increase in total lung capacity, however this is not the case. How do the authors explain this?
- In the univariate analysis, there is no significant association between muscle parameters and sex difference despite the literature (and this is contradictory with the sentence in the discussion page 13 “In contrast, females exhibited a more pronounced decrease in SMA, SMI, and SMD than males» could the authors discuss the lack of significance and also tone down their conclusion as there is no statistically significant difference?
- Discussion, 1st paragraph: “COPD. We observed a general decline in body composition measurements, except for an increase in IMFD” again, please tone down this conclusion as there was no statistically significant difference.
- Discussion, 2nd paragraph: “A plausible explanation for the increase seen in our study could be the result of more effective control of lung hyperinflation attributed to rehabilitative measures combined with fewer exacerbations/years.”: such information is indeed very important/relevant for the results. It might also account for the decrease in muscular fat. Could the authors provide the percentage of patients undergoing pulmonary rehabilitation during the period of follow-up?
- Discussion, 3d paragraph: “As anticipated, we observed a decline in skeletal muscle measurements, consistent” as previously mentioned, please tone down this statement (e.g., trend towards decline)
- Finally, the decrease in muscular fat area associated to an increase in muscular fat density seems of particular interest to me. Do the authors think that it might represent of form of tissue remodelling (fat “compaction”)? Has this already been described in the literature? I think it would be a valuable point to discuss.
Minor comments:
- Introduction, 2nd paragraph: what do the authors mean by “low and reduced muscle mass”?
- Introduction, end of 2nd paragraph: references on increased mortality associated with skeletal muscle wasting in COPD could be added (e.g. Marquis AJRCCM 2002, Schols Am. J. Clin. Nutr. 2005, Swallow Thorax 2007…)
- Introduction, 3d paragraph, first sentence: I believe other factors are implicated (e.g. oxidative stress, genetic&epigenetic factors, etc), this could be added to the sentence with appropriate references as well (e.g. the 2014 ATS/ERS statement).
- Methods, Figure1: what are the correspondences for the colors purple and orange?
- Results: Table 3 presents the results of the univariate linear regression, but only age is associated with muscle parameters (and this will be later taken into account in the multivariate analysis). I suggest this Table to be moved to the appendix in order to improve clarity and highlight the main message.
- - Discussion, 3d paragraph: “As anticipated, we observed a decline in skeletal muscle measurements” but this decline was only a trend: I suggest the authors tone down their conclusions in this paragraph. Probably the short duration of follow-up, the relatively small ample size and the lack of stratification by sarcopenic status contribute to the lack of significance.
Typos:
- Introduction: the same sentence is written twice (see end of 3d paragraph and 4th paragraph)
- Introduction: “lunge” volumes in the 3d objective
Reviewer 3 Report
Comments and Suggestions for Authors
The area of investigation in the present study is interesting and within the confines of the study parameters the clinical assessments are robust. I only have what i consider to be minor questions.
1) can the timing of the study please be indicated. If any part of this study was during, or around the COVID restrictions (which I am uncertain if they occurred in the country in which this study occurred) it could contribute to changed fat mass independent of COPD?
2) are any exercise capacity (eg 6 minute walk test) data available?
3) was there any change in medication over the study period (i.e. high dose steroids)?
4) I doubt the data would be available, but are there data from people without COPD?
5) if any more complex / investigational lung function measurements were made (eg DLCO, FOT ect) can these please be included.
Reviewer 4 Report
Comments and Suggestions for Authors
Subject of this manuscript is very interesting and to know if loss muscle mass could be predicted in COPD is in my opinion very important. The manuscript is well elaborated and is a good scientific exercise with data that not always are valued in respiratory diseases.
I have some concerns or questions:
1) Methodology - The fact of not have assessed the levels of physical activity is a major problrm due to the relationship between exercise and muscle mass and respiratory diseases and the ffort and studies that are made in that subject. I believe that at least some of the participants participate in some kind of programms of clinical exercise. Results could be biased by that.
2) Number of participants is not enough (in my oppinion) to make a relationship such as this manuscript try to do. Why do not make a longitudinal study since the first assessment (prospective) instead of take advantage of longitudinal but retrospective data ?
3) Results - Why have heigth changed (increased?) in participants over 60 years. Not normal. So, comparing heigth between baseline and 12-months follow-up and find a value of "p" is not...scientific!
4) Could you discuss the changes in weigth betwen baseline and 12-months follow-up? Age and disease could impact the body composition components? If yes how
5) Table 1 is divided intwo pages which makes it difficult to read since the titles (subject - COPD-stage) is not in the same page. Try to make it more "reader-friendly".
6) In the first paragraph after the tabnle 1, authors say that all skeletal muscle components decrease during the time assessed. But that was the expected with ageing and medication. Not the relationship with pulmonary function assessed by plethismography. And in these observations i feel the necessity of levels of physical activity.
7) These 35 patients could be analised controlling for levels of physical activity (or at least sedentary vs active). In my opinion without this is not accurate trying to establish a relationship between longitudinaly changes in body composition components and components of pulmonary function that by the way it seems not decrease in one year.
It will be fantastic if we could predict changes in pulmonary function by skeletal mass changes but in my oppinion this is not possible with only 35 patients and without physical activity reference. Maybe if this is considered a "pilot study".
Round 2
Reviewer 2 Report
Comments and Suggestions for Authors
Thank you for responding to all my comments.
Author Response
Dear reviewer,
We appreciate your swift and positive feedback on the revisions we submitted. It's truly gratifying to know that our efforts to address your comments have met your expectations.
On behalf of the author team
Warm regards
Reviewer 4 Report
Comments and Suggestions for Authors
To the authors
Thank you for having adressed all my concerns on your study, even some of them could not be changed...). I am happy that you recognize your limitations (limitations of the study) since that help readers to understand that this is only a pilot study.
With your corrections and explanations i agree accept this manuscript
Author Response
Dear Reviewer,
Thank you for your thoughtful feedback and for recognizing the efforts made to address the concerns raised, including the study's limitations. We are grateful for your understanding and happy to learn that the manuscript is considered acceptable for publication.
Your guidance has been invaluable in enhancing the quality and clarity of our work. We look forward to contributing to the journal and the broader scientific dialogue.
Warm regards,
The author team